# Intermolecular Non-Covalent Carbon-Bonding Interactions with Methyl Groups: A CSD, PDB and DFT Study

**DOI:** 10.3390/molecules24183370

**Published:** 2019-09-16

**Authors:** Tiddo J. Mooibroek

**Affiliations:** van ’t Hoff Institute for Molecular Sciences, Universiteit van Amsterdam, Science Park 904, 1098 XH, Amsterdam, The Netherlands; t.j.mooibroek@uva.nl; Tel.: +31(0)205-25-72-08

**Keywords:** intermolecular interactions, non-covalent interactions, carbon-bonding interactions, crystal structure database analysis, density functional theory

## Abstract

A systematic evaluation of the CSD and the PDB in conjunction with DFT calculations reveal that non-covalent Carbon-bonding interactions with X–CH_3_ can be weakly directional in the solid state (*P* ≤ 1.5) when X = N or O. This is comparable to very weak CH hydrogen bonding interactions and is in line with the weak interaction energies calculated (≤ –1.5 kcal·mol^−1^) of typical charge neutral adducts such as [Me_3_N-CH_3_···OH_2_] (**2a**). The interaction energy is enhanced to ≤–5 kcal·mol^−1^ when X is more electron withdrawing such as in [O_2_N-CH_3_··O=C^dme^] (**20b**) and to ≤18 kcal·mol^−1^ in cationic species like [Me_3_O^+^-CH_3_···OH_2_]^+^ (**8a**).

## 1. Introduction

The manner in which molecules interact with one another is largely determined by non-covalent interactions.c [1] So-called ‘σ-Hole interactions’c [2,3,4,5] like hydrogen bonding are prominent identifiable interactions that bear biological significance [6]. Such σ-hole interactions have also been identified with other non-metals [7,8,9,10,11] like halogen atoms to generate halogen bonding interaction [12,13]. The impact of halogen bonding interactions on molecular biology has come into focus since about 2004 [14]. Indeed, evaluations of the protein data bank (PDB) [15] revealed that halogen bonding is structurally very similar to hydrogen bonding [12,14,16,17,18] and can be functionally relevant [19,20,21,22]. Relatively weak π-hole interactions [4,23,24,25,26,27,28,29,30,31] involving organic carbonyls, [26,32,33,34,35,36] π-acidic aromatics, [37,38] metal carbonyls [33,34,36,39] and nitro-compounds [40,41,42,43,44,45] are increasingly acknowledged as relevant drivers of molecular aggregation such as in ligand-protein complexes.

The impact of a novel type of weak interaction on molecular recognition phenomena naturally leads one to speculate that other non-canonical interactions may play a similar role. One interesting candidate are σ-hole interactions involving sp^3^-hybridized C-atoms. Such interactions have been studied since about 2013 [7,46] and are particularly interesting because sp^3^-C is abundant in living systems. More specifically, the methyl group (X–CH_3_, where X = any atom or group) is frequently encountered in natural and synthetic compounds and ‘non-covalent Carbon bonding’ involving methyl groups has thus been studied by various researchers [47,48,49,50,51,52,53,54,55,56,57,58,59]. Most of these contributions are computational inquiries, while a small amount of these articles also deals with an analysis of non-covalent Carbon bonding interactions in protein structures present in the Protein Data Bank (PDB) [47,50,56]. Interestingly, none of the studies so far have systematically evaluated the crystal structure data present in the Cambridge Structure Database (CSD) [60,61]. What is more, evaluations of the PDB were largely anecdotal or only considered structures that comply to the (rather strict) geometric criteria of a Carbon bonding geometry. Some also included *intra*molecular contact distances (which are notoriously difficult to evaluate).

In this contribution a combined CSD and PDB evaluation is presented aimed at elucidating whether electron rich entities have a preferential orientation around a methyl group within a rather large envelope, i.e., whether intermolecular non-covalent Carbon bonding interactions with methyl groups are directional. For evaluative purposes, several Density Functional Theory (DFT) computations were conducted as well. This combined database/DFT study reaffirms that non-covalent Carbon-bonding interactions with X-CH_3_ can be significant, although the interaction is hardly directional, in particular when the methyl group is poorly polarized such as most C–CH_3_ structures.

## 2. Materials and Methods

### 2.1. General Information on Database Analyses

The CSD [60,61] version 5.40 including two updates (until May 2019) was inspected using ConQuest [62] version 2.0.2 (build 246353, 2019). X-ray powder structures were omitted from the searches, which were further limited to structures containing 3D coordinates and those with an R-factor ≤ 0.1. The PDB was queried using Relibase [63] 3.2.3 and restricted to protein and DNA crystal structures where the packing environment was also searched. No other restrictions were imposed on the PDB search. Datasets were obtained using the general query shown in Figure 1a. The methyl groups were split in those connected to a C, N, O, P, or S atom (X in the figure, in the PDB search specified as part of a ligand). The interacting ‘electron rich’ partners (ElR in the figure) considered were a water, amide or carboxy-O atom, a sulphur atom or the centroid of an aryl ring (in the PDB search always specified as part of the protein). The geometric constraints imposed on the searches were that the *inter*molecular distance ***d*** between the methyl C-atom and ElR was ≤5 Å and that the X–CH_3_∙∙∙ElR angle (α) was 90°–180°. All the data were thus confined within a hemisphere with a basal radius of 5 Å, centered on the methyl C-atoms as is shown in Figure 1b.

### 2.2. Methodology to Generate P(α) Plots

The datasets obtained as described above (2.1) were analysed to assess whether the distribution of ElR within the methyl-centered hemisphere reflects any directionality. This method has been successfully applied to assess the directional behaviors (in the solid state) of various other weak non-covalent interactions such as anion/lone-pair-π, [29,64] CH-π, [11,65] halogen-π [66,67] and nitro π-hole interactions [42,68]. The method works by first computing the freely accessible volumes at each α-value (α^free^) by subtracting the volume of a model methyl group from a spherical cone with 5 Å height and a cone angle of 180-α. This can be achieved by using the 3D-drawing program Autodesk^®^ Inventor^®^ Pro [29]. This is illustrated in Figure 1b, where the spherical cones are shown at 10° intervals. The model methyl group was generated by using standard aliphatic C–H bond distances (1.06 Å) [69] and the van der Waals radius of C (1.70 Å) and H (1.09 Å) [70]. The interfering volume between each spherical cone and the model methyl group can be obtained using the ‘inspect interference’ option in Autodesk^®^ Inventor^®^ Pro; the red part in Figure 1b is the interfering volume involving a spherical cone with a cone angle of 60° (i.e., at α = 120°). The volume differences between such ‘free’ volumes with increasing values of α thus give the absolute volume distribution of freely accessible volume around a methyl model within the hemisphere, as a function of α: Δα^free^(α). Dividing each volume (Δα^free^) in this distribution by the total freely accessible volume (i.e., the volume of a hemisphere minus the interfering volume of the model methyl group in that hemisphere) thus gives the relative volume distribution as a function of α: Δ^rel^α^free^(α). This distribution is the random (or volume) distribution. The data retrieved form the CSD and the PDB can be binned as a function of α. Relating this binned data to all the data in a dataset thus gives the observed relative distribution as a function of α: Δ^rel^α^data^(α). The quotient of this relative data distribution over the random distribution is a measure for the actual probability (*P*) of finding data at a certain value of α. That is, *P*(α) is unity for a random distribution of data, while *P*-values larger than unity reveal a relative concentration of data, which is indicative of attractive interactions.

### 2.3. Methodology to Generate N(**d’**) Plots

A second analysis involved plotting the hit fraction (*N*) for a subset with α = 160°–180° as a function of the van der Waals corrected H_3_C∙∙∙ElR distance ***d’*** (i.e., ***d*** – vdW(C) – vdW(ElR)): [70] *N*(***d’***). Such distributions show how much of the data is involved in van der Waals overlap with the methyl C-atom along the vector of the X–CH_3_ bond and how such data is distributed. For attractive interacting pairs this distribution is expected to exhibit a peak-like feature, or an S-like curvature when the cumulative hit fraction is used.

### 2.4. Computational Methods

DFT geometry optimization calculations were performed with Spartan 2016 at the B3LYP [71,72]-D3 [73]/def2-TZVP [74,75] level of theory, which is known to give accurate results at reasonable computational cost and a very low basis set superposition error (BSSE) [74,75]. The typical starting geometry for possible Carbon bonding adducts was set to ***d’*** = −0.1 Å and α = 180°, and in the case of dimethylacetamide the C···O=C angle was also set to 180°. The geometry optimizations were performed without any constraints. For other geometries (e.g., a H-bonded geometry), the molecular fragments were manually oriented in a suitable constellation before starting an unconstrained geometry optimization. The Amsterdam Density Functional (ADF) [76] modelling suite at the B3LYP [71,72]-D3 [73]/TZ2P [74,75] level of theory (no frozen cores) was used for energy decomposition and ‘atoms in molecules’ [77] analyses. Details of the Morokuma-Ziegler inspired energy decomposition scheme used in the ADF-suite have been reported elsewhere [76,78] and the scheme has proven useful to evaluate hydrogen bonding interactions [79].

## 3. Results and Discussion

### 3.1. P(α) Plots

A numerical overview of the amount of crystallographic information files (CIFs) and protein data bank files (PDBs) for each search query is given in Appendix A, together with the amount of hits found in each dataset (a .cif or .pdb file can contain multiple hits). Shown in Figure 2 are the *P*(α) plots for the CSD (left) and PDB (right) data plotted at 5° intervals involving X = C, N, O and ElR = water-O (top) or amide-O (bottom). These datasets were chosen because they all contained a large number of hits (>7,500) and thus allow for the most reliable comparison. A complete set of *P*(α) plots is provided in Appendix A.

The data plotted in Figure 1 largely trace the line at *P* = 1 (highlighted in green), which is indicative of a random distribution of data. For X = O and N, these values are somewhat above unity around α = 160°–180° for water O-atoms in both databases and for amide O-atoms in the CSD. The maximum *P*-values are very small at about 1.5, which indicates a very small amount of directionality. Indeed, maximum *P*-values for several weak inter-molecular interactions are: ~2.5 for CH-π; [11,65] ~3 for π interactions with nitro compounds; [42,68] ~2.5–5 for anion-π and lone-pair-π; [29,64] ~2.5–10 for halogen-π [67] and also about 2.5–10 for halogen bonding with aryl-halogens. [66] Interestingly, the *P*-values did not peak near α = 90°–120°, an angle congruent with hydrogen bonding. These data thus suggest that the Carbon binding geometry is more directional than a hydrogen bonding geometry, although this directionality is very weak. In all cases for X = C, the *P*-values around α = 160°–180° are below unity, suggesting that the Carbon bonding geometry is least favored in these instances. The data for ElR = RCO_2_ and R_y_CS are very similarly distributed and the data for aryl rings is skewed towards α = 160°–180° only for the CSD data (for X = C, N, O, P and S, see Appendix A). For all other cases where X = P, very few hits were obtained (most numerous was RCO_2_ in the CSD with *N* = 1,454). While some datasets with X = S were of a reasonable size, too many were well below *N* = 7,500 and these data will thus not be discussed in the main text (there is a small discussion in the caption of Appendix A). 

### 3.2. N(**d’**) plots

The data characterized by α = 160°–180° was inspected further by means of *N*(***d’***) plots as described in the methods and materials section. A numerical overview of the amount of data in each dataset as well as the (relative) amount of van der Waals overlap found in each dataset is given in Appendix A. Shown in Figure 3 are plots of the hit fractions (in %) as a function of ***d’*** (the van der Waals corrected ***d***) for X = C, N, O and ElR = water or amide. The same data plotted as cumulative hit fractions is shown in Appendix A and the *N*(***d’***) plots for all datasets containing >500 hits are shown in Appendix A.

In the CSD (left), The data involving N/O–CH_3_∙∙∙O^amides^ are very similarly distributed and grouped near ***d’*** ≈ 0 Å with about 30% of all the data involved in van der Waals overlap. Likewise, data involving N/O–CH_3_∙∙∙O^waters^ are also very similarly distributed but group near the larger ***d’*** ≈ 0.25 Å with 15% van der Waals overlap. The C–CH_3_∙∙∙O^amides/waters^ data hardly displays van der Waals overlap (3–4%) and is broadly grouped around ***d’*** = 0.4 Å for waters and not grouped at all for amides. Similar trends are present in the PDB (right), albeit the features are much less pronounced and the N/O–CH_3_∙∙∙O datasets with amides and waters are very similar. 

The data presented in Figure 3 thus imply a somewhat directional nature of X–CH_3_∙∙∙O^amides/waters^ interactions for X = N and O, but not at all for X = C. These findings are in line with the lack of directionality observed in the *P*(α) plots for X = C and the somewhat directional behavior for X = N or O (see Figure 2). A likely explanation for this is the larger (Pauling) electronegativity of N (3.04) and O (3.44) compared to C (2.55), resulting in a larger degree of polarization of the X–CH_3_ bond for X = N or O. Another conceivable manner to make a methyl group pore electropositive is to bind it to a cationic fragment such as in protonated or quaternary R_3_N^+^–CH_3_ fragments. Thus, an additional dataset was retrieved from the CSD involving R_3_N^+^–CH_3_···ElR pairs that fulfilled the ***d*** ≤ 5 Å and α = 160°–180° criteria (see bottom entries in Appendix A). Shown in Figure 4 are the *N*(***d’***) plots of the most numerous datasets involving R_3_N^+^–CH_3_ (hexagonals, N^+^) together with similar datasets involving all possible N–CH_3_ fragments (diamonds, N). In all three cases (ElR = water, carboxy or aryl), the distribution is shifted to lower ***d’*** distances for cationic R_3_N^+^–CH_3_, which means that relatively more van der Waals overlap is present in these datasets. As can be seen especially from the cumulative *N*(***d’***) plots (left), the grouping is tightest with carboxy O-atoms (green), followed by water O-atoms (red) and aryl rings centroids (grey) are not grouped at all (nearly linear).

### 3.3. Computations

In order to gain insight into the nature and energetics of possible non-covalent Carbon bonding interactions involving various methyl groups and water or amide O-atoms, DFT calculations were performed of X–CH_3_ adducts with water and with dimethylacetamide (dma, see methods section for details). An overview of these adducts is given in Table 1, together with α of the optimized structures, the total interaction energy of the adducts in kcal·mol^−1^ and the percentages of electrostatic (E), orbital (O), and dispersion (D) interactions that contribute to this total energy [79]. Perspective views and atoms-in-molecules analyses of all converged structures are shown in Appendix A and several representative examples are shown in Figure 5.

For comparison purposes, adducts with ethane were computed as shown in entries 1 of Table 1. Both structures converged to a hydrogen bonding interaction. ΔE = −1.2 kcal·mol^−1^ in **1a** and a methyl acts as electron donating site; i.e., an O–H···C hydrogen bonding interaction. This can be understood due to the polarization in ethane, where both C’s are most electronegative and are polarized by the H-atoms. Adduct **1b** is about twice as stable with ΔE = −2.8 kcal·mol^−1^ and also features a hydrogen bonding interaction, but now between a methyl CH and a π-bond in dma. Both **1a** and **1b** are stabilized mainly by dispersion (46–58%), then electrostatics (32–26%) and least by orbital interactions (22–16%). The neutral water adducts where X = permethylated N, O, P, or S are energetically nearly identical to the ethane adduct (**2a**–**5a**). Like with ethane, **4a** converged in a O–H···C hydrogen bonding interaction, which can be rationalized by the lower (Pauling) electronegativity of P (2.19) compared to C (2.55). **2a**, **3a** and **5a** converged at a geometry consistent with a Carbon bonding interaction. This is illustrated for structure **3a** in Figure 5, where a single bond critical point (bcp) is located between C and O with a bond density of 0.60 · 10^2^ a.u.. This can be rationalized by the higher (Pauling) electronegativity of N (3.05), O (3.44) and S (2.58) compared to C (2.55). Electrostatics or dispersion are the main energetic stabilizing factor in adducts **2a**–**5a** which is typical for weak and non-directional interactions like in adduct **1a**. A similar series with dma was computed as adducts **2b**–**5b**. Only **2b** converged in a geometry consistent with a Carbon bonding interaction (with dispersion as main driver) while the others are C–H···O hydrogen bonding interactions. Carbon bonding interactions with regular permethylated main group elements are thus comparable to very weak C–H hydrogen bonding interactions and less than about −1.5 kcal·mol^−1^ in strength. These energies are in line with earlier computations with the adducts [H_2_N-CH_3_···OCH_2_] [47] and [HO-CH_3_···OH_2_] [48] of −0.7 and −1.0 kcal·mol^-1^ respectively.

The cationic adducts **6**–**11a** were computed as well and the most stable of these involved the Me_3_C^+^ carbocation in **6** (adducts with pentamethylated Carbon are unstable). The bonding interaction in **6b** is largely covalent, as evidenced by the interaction energy of −82.4 kcal·mol^−1^, the large orbital contribution (55%), a dense bond critical point (18.4 · 10^2^ a.u.) and a clear pyramidalization of the central C-atom (see Appendix A). Of the other adducts, all except **6a** (Me_3_C^+^···O interaction) and **9** (with the least electronegative P) converged into an X–CH_3_···O Carbon bonding geometry. This is illustrated for **8a** and **11a** in Figure 4, where a clear bcp can be seen in between ^methyl^C and O^water^ with a bond density of 1.15 · 10^2^ and 0.98 · 10^2^ a.u. for **8a** and **11a** respectively. The bonding energies in **7**–**11a** are mainly electrostatic in origin (~70%) and about −8 kcal·mol^−1^ for water and −15 kcal·mol^−1^ for dma. The most stable adducts in both series involved the most electronegative O (3.44) in Me_3_O^+^ (**8**). Two alternative configurations with *N*-methylpyridinium were also computed (**12** and **13**). In **12**, the O points in between two CH hydrogens as is illustrated for **12a** in Figure 4. In adducts **13** the O atom is located directly above the cationic N^+^. Both **12** and **13** are more stable than the Carbon bonding geometry found in **11**, suggesting that hydrogen bonding interactions are most preferred. The interaction energies of Carbon bonding interaction with cationic species is similar to previous data of the adducts: [H_3_N^+^-CH_3_···OCH_2_] (−9.7 kcal·mol^−1^); [47] [Me_3_N^+^-CH_3_···OC(H)NH_2_] (−13 kcal·mol^−1^); [49] [Me_2_S^+^-CH_3_···OH_2_/NH_3_/OCH_2_] (about −8–9 kcal·mol^−1^); [47,49] and [R_2_S^+^-CH_3_···various lone-pairs] (about −9.0 kcal·mol^−1^) [50]. 

As the calculations with cationic species imply that electron withdrawing substituents amplify the Carbon bonding interaction, it was decided to compute adducts with small molecules that have an electron withdrawing group: Iodomethane (**14**), 1,1,1-trifluoroethane (**16**), acetonitrile (**18**) and nitromethane (**20**). All these adducts converged as a Carbon bonding geometry and are energetically favorable by 1.6–2.8 kcal·mol^−1^ for water and 3.0–4.9 kcal·mol^−1^ for dma. The adducts involving nitromethane (**20**) were most stable and are shown in Figure 5, together with an aim analysis revealing a single C···O bcp (*ρ* = 0.79 · 10^2^ and 0.83 · 10^2^ a.u. for **20a** and **20a** respectively). For the water adducts, H-bonding geometries were also optimized: **15a**, **17a**, **19a** and **21a**. All these adducts were about twice as stable and the Carbon bonding geometry. This can be ascribed to the fact that besides a C–H···O hydrogen bonding interaction, another weak hydrogen bonding interaction is present as well (i.e., C–H···I in **15a**, C–H···F in **17a** and C–H···π in **19a**, see also see also Appendix A). For example, a dimer of HCF_3_ is estimated at −2.6 kcal·mol^−1^ and exhibits two weak C–H···F hydrogen bonding interactions (not shown). The interaction energies with neutral yet polarized methyl groups (**14**–**20**) liken those reported by others for: [F-CH_3_···OH_2_/NH_3_/PH_3_] (about −2–3 kcal·mol^−1^); [48,51,52,53] [F-CH_3_···C_2_H_2_] (−1.2 kcal·mol^−1^); [54] [Hlg-CH_3_···C_2_H_4_/NH_3_/PH_3_] (−1–4 kcal·mol^−1^); [53,55] [NC-CH_3_···C_2_H_4_/dma] (−3–5 kcal·mol^−1^); [55,56] and [O_2_N-CH_3_···dma] (−4.9 kcal·mol^−1^) [56].

### 3.4. General Discussion

From all the calculations collected in Table 1 it is evident that the interaction energies of Carbon bonding geometries with the sp^2^-O in dma (adducts ‘b’) is consistently about twice as strong as the interaction with sp^3^-O in water (adducts ‘a’). This is in line with the larger amount of van der Waals overlap observed in the *N*(***d’***) plots (Figure 2, ~30% for amides vs ~15% for water). The interaction energies of adducts with a Carbon bonding geometry range from very weak (below −1.5 kcal·mol^−1^ in **2**, **3**, **5**), to moderately weak (between −1.5 and −5 kcal·mol^−1^ in **14**, **16**, **18** and **20**) to fairly strong in the cationic adducts (between −7 and −18 kcal·mol^−1^ in **6**–**11**). ΔE becomes smaller (more stable) in the order **2** < **14** < **16** < **18** < **20** < **7** < **8**. Within this series, the orbital contribution remains constant at about 15–20%, while the electrostatic component increases from 30–35% in **2** to about 65% in **8**. This implies that stronger Carbon bonding interactions are mainly driven by electrostatic interactions and that weaker such adducts are driven by dispersion. These computational results are consistent with recent literature reports [47,48,49,50,51,52,53,54,55,56,57,58,59] and the database analyses presented here; neutral adducts are very weak and thus hardly (or non) directional but can be made stronger (and thus presumably more directional) when X in X–CH_3_ is strongly polarized (see especially Figure 4). The relevance of Carbon bonding interactions with methyl groups is thus likely limited to highly polarized and/or cationic species. While this limits the scope considerably, it is worth pointing out that ligands with methyl groups related to those in adducts **2**–**10** and **14**–**21** are abundant within proteins structures and that cationic methyl groups also occur. For example, methylated methionine residues in methyl transferases [80] and nicotinamide derivatives such as nicotinamide adenine dinucleotide [81,82].

## 4. Summary and Conclusions

The CSD and the PDB were systematically evaluated for potential directional behavior of intermolecular non-covalent Carbon bonding interactions involving X–CH_3_ and electron rich entities such as O/S atoms or an aryl ring (ElR) within a hemisphere of 5 Å basal radius (centered on C). It was found that X–CH_3_···ElR interactions can be as directional as very weak hydrogen bonding interaction involving C–H (*P*_max_ ≤ 1.50) but not directional at all when X = C. Grouping of data with significant amounts of van der Waals overlap (up to ~30%) was observed in various sub-datasets in the region where the X–CH_3_···ElR angle α is 160°–180°. These distributions were significantly shifted to shorter distances (i.e., more van der Waals overlap) in the case of cationic R_3_N^+^–CH_3_···O^water/amide^ compared to charge-neutral R_2_N–CH_3_···O^water/amide^ interactions.

Model DFT calculations revealed that charge neutral X–CH_3_···O adducts with water and dimethylacetamide are very weak (≤ –1.5 kcal·mol^−1^ in **2**, **3a**, **5a**) and are often not the energy minima of the adducts (**1**, **3b**, **4**, **5b**). The interaction energies can be increased by deploying a more electron withdrawing X (–1.5 to –5 kcal·mol^−1^ in **14**, **16**, **18** and **20**). Rendering X cationic leads to even more stable adducts (–7.0 to –18 kcal·mol^−1^) in **7**, **8**, **10** and **11**). Carbon-bonding adducts with dimethylacetamide are consistently twice as stable as those with water. Energy decomposition analyses showed that increased stability is driven by electrostatics and atom-in-molecule analyses regularly gave a clear bond critical point involving the methyl C-atom.

It is thus concluded that this combined database / DFT study reaffirms that intermolecular non-covalent Carbon interactions with X–CH_3_ is electrostatically driven and can be significant. The interaction can even by mildly directional in the solid state (comparable to weak CH hydrogen bonding interactions), provided X is sufficiently electron withdrawing.

## Figures and Tables

**Figure 1 molecules-24-03370-f001:**
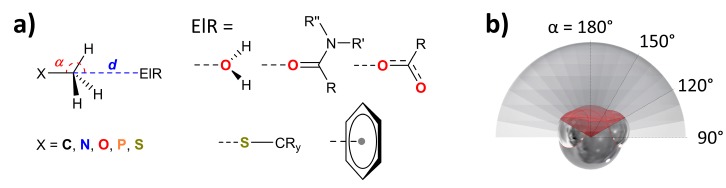
Representation of the method used to retrieve and analyse data from the CSD and the PDB. (**a**) general query to obtain data with ***d*** ≤ 5Å, α = 90°–180°, X = C, N, O, P or S and ElR (electron rich entity) is as indicated. (**b**) Illustration of the method used to assess directionality (see text for details).

**Figure 2 molecules-24-03370-f002:**
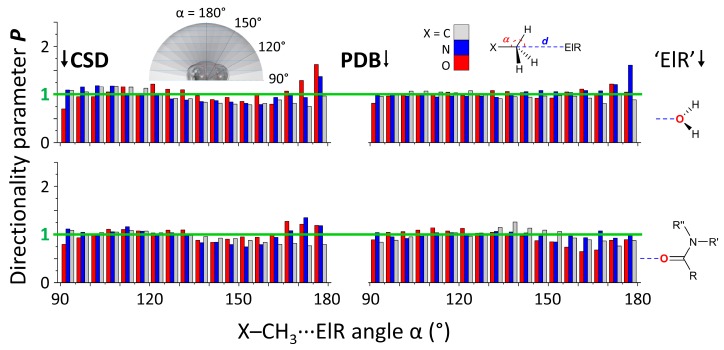
*P* (α) directionality plots for the data retrieved from the CSD (left) and the PDB (right) using the general query shown in the top-right inset figure for X–CH_3_∙∙∙ElR pairs. X can be C, N or O and ‘ElR’ can be a water or an amide O-atom. The insert figure in the top left is intended as a guide to the eye to interpret the spatial location of data with a certain value of α. Due to the amount of data per dataset (see Appendix A for numerical overview), the plots are given at a 5° resolution for α. A full set of *P*(α) plots (i.e., for all the X *vs* ElR pairs in Figure 1) is given in Appendix A. The *P* value of 1 is highlighted in green and indicates a random distribution of data. *N* (CSD/PDB) = 46,000/29,508 (C, water); 18,170/17,101 (N, water); 7190/11,392 (O, water); 53,473/22,538 (C, amide); 7663/10,855 (N, amide); 9,158/11,064 (O, amide).

**Figure 3 molecules-24-03370-f003:**
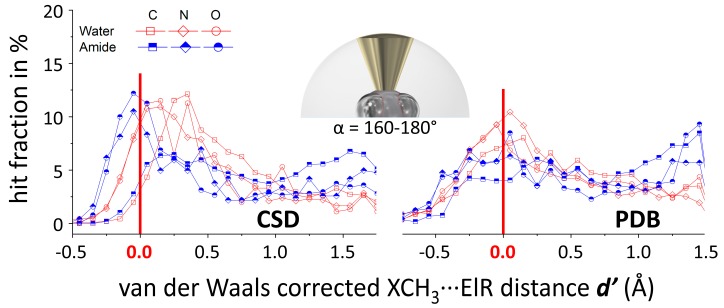
Hit fraction (in %) as a function of the van der Waals corrected XCH_3_···ElR distance ***d’*** (in Å) for several datasets from Appendix A (*α* ≥ 160°, as illustrated by the inset figure). The interacting pairs involve water-O (red, empty) or amide-O (blue, half-filled) with X = C (squares), N (diamonds), or O (circles). See Appendix A for the same data plotted as a cumulative hit fraction. *N* (CSD/PDB) = 2,376/1,622 (C, water); 1,089/1,186 (N, water); 452/757 (O, water); 2,599/1,179 (C, amide); 504/650 (N, amide); 640/483 (O, amide).

**Figure 4 molecules-24-03370-f004:**
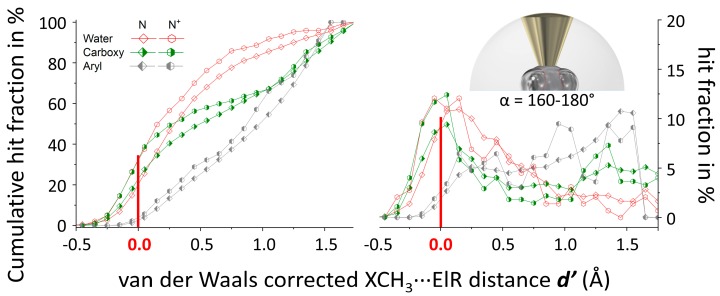
Cumulative (**left**) and regular (**right**) hit fraction (in %) as a function of the van der Waals corrected XCH_3_···ElR distance ***d’*** (in Å) for the datasets from Appendix A (*α* = 160°–180° as illustrated in the inset figure). The interacting pairs involve water-O (red, empty), carboxy-O (green, right-filled) or the centroid of an aryl ring (grey, left-filled). X can be any N (diamond) or a cationic (tetravalent) N^+^ (hexagonal). *N* (N/N^+^) = 1,809/290 (water, red); 2,062/274 (carboxy, green); 10,585/527 (aryl, grey).

**Figure 5 molecules-24-03370-f005:**
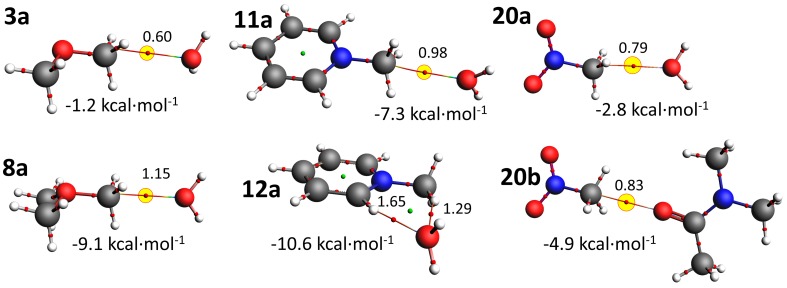
Ball and stick representations of molecular adducts selected from Table 1 that were optimized by DFT (B3LYP-D3/def2-TZVP). The thin lines are bond paths (bp’s) and the small red spheres are bond critical points (bcp’s) obtained from an ‘atoms-in-molecules’ analysis. The bond density (*ρ*) is in arbitrary units ·10^2^ and bcp’s indicative of non-covalent Carbon bonding have been highlighted in yellow.

**Table 1 molecules-24-03370-t001:** Numerical overview of adducts computed with DFT between an indicated X-CH_3_ methyls and water (adducts ‘a’) or dimethylacetamide (dma, adducts ‘b’). Using ADF at the B3LYP-D3/def2-TZVP level of theory, interaction energies (in kcal·mol^−1^) were computed and an energy decomposition analyses is shown as a percentage of the total amount of interaction energies split up as electrostatic (E), orbital (O) and dispersion (D) interactions. Entries in grey are not consistent with a Carbon bonding geometry. See Appendix A for perspective views and atoms in molecules analyses.

Interacting X–CH_3_:	Water–O				Dimethylacetamide (dma) – C=O			
	adduct	α (°)	ΔE	E/O/D in %	adduct	α (°)	dE	E/O/D in %
**Me–CH_3_**	**1a**	170^a^	−1.2	32/22/46%	**1b**	122^f^	−2.8	26/16/58%
**Me_2_N–CH_3_**	**2a**	174^b^	−0.8	36/18/45%	**2b**	173^b^	−1.6	33/22/44%
**MeO–CH_3_**	**3a**	175^ b^	−1.2	48/17/35%	**3b**	164^f^	−3.2	27/16/56%
**Me_2_P–CH_3_**	**4a**	167^a^	−1.2	28/32/41%	**4b**	91^d^	−4.2	45/21/33%
**MeS–CH_3_**	**5a**	177^ b^	−1.0	44/17/38%	**5b**	100^d^	−3.7	46/21/32%
**Me_2_C^+^–CH_3_**	**6a**	75^c^	−12.6	66/22/12%	**6b**	109^c^	−82.4	43/55/2%
**Me_3_N^+^–CH_3_**	**7a**	173^b^	−7.8	73/15/12%	**7b**	170^b^	−14.7	70/19/10%
**Me_2_O^+^–CH_3_**	**8a**	176^b^	−9.1	72/18/10%	**8b**	175^b^	−17.6	69/22/8%
**Me_3_P^+^–CH_3_**	**9a**	55^d^	−8.2	69/14/18%	**9b**	71^d^	−21.7	64/23/13%
**Me_2_S^+^–CH_3_**	**10a**	168^b^	−7.6	73/15/12%	**10a**	170^b^	−15.0	70/20/10%
**^Py^N^+^–CH_3_**	**11a**	170^b^	−7.3	73/15/12%	**11b**	175^b^	−14.2	71/19/10%
	**12a**	93^d^	−10.6	70/22/9%	**12b**	91^d^	−20.3	65/27/8%
	**13a**	68^e^	−9.0	70/14/16%	**13b**	74^e^	−18.2	64/21/15%
**I–CH_3_^g^**	**14a**	175^b^	−1.6	53/18/28%	**14b**	175^b^	−3.0	52/20/28%
	**15a**	84^d^	−3.8	55/23/21%				
**F_3_C–CH_3_^ g^**	**16a**	179^b^	−1.8	59/13/27%	**16b**	175^b^	−3.4	59/15/26%
	**17a**	76^d^	−3.8	60/15/24%				
**N≡C–CH_3_^ g^**	**18a**	158^b^	−2.3	65/13/23%	**18b**	172^b^	−4.3	62/15/23%
	**19a**	68^d^	−5.1	63/19/18%				
**O_2_N–CH_3_^ g^**	**20a**	165^b^	−2.8	66/13/21%	**20b**	179^b^	−4.9	64/15/21%
	**21a**	86^d^	−6.4	63/25/12%				

^a^ One of the water H-atoms and one of the RCH_3_ C-atoms are closest to each other; ^b^ Carbon bonding interaction geometry; ^c^Interaction with the cationic C; ^d^ Hydrogen bonding interaction(s); ^e^ Interaction with cationic N; ^f^ CH-π interaction; ^g^ The interaction energy with benzene was also computed, starting from a geometry with X–CH_3_···benzene centroid = 180°. All adducts converged at a geometry where this angle was about 90° degrees. Interaction energies are about 4–5 kcal·mol^−1^ and dominated by electrostatics (35–40%) and dispersion (40–50%, see Appendix A for details).

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
