# Peer review of "Intermolecular Non-Covalent Carbon-Bonding Interactions with Methyl Groups: A CSD, PDB and DFT Study"

_molecules, 2019, doi:10.3390/molecules24183370_

Round 1

Reviewer 1 Report

1> The introduction should be improved. Not enough background is given to the research background. 

2> The language needs improvement in many cases. 

3> Why is the table called Table S1 and not Table 1? 

4>  There needs to be more improvement in the conclusion and abstract. 

Author Response

I am grateful for this reviewer’s time and suggestions to which I have responded below:

vs 1: Personally I like a short introduction, but please note that the paper in its present form already contains more than 80 references and that about 55 of these appear solely in the introduction. In my opinion, there is no need for more information on the background and I hope that this referee and the editor will respect my opinion (which apparently is shared by referee 2).

vs 2: Many thanks for the comment; I indeed found some typographical errors on reading the manuscript again and thus fixed them (highlighted in yellow). It is my hope that the editorial process will root out possible other lingual issues.

vs 3: Many thanks for pointing out this typographical error; Table S1 in the main text should indeed be Table 1. This has been changed in the manuscript.

vs 4: I have expanded the abstract and the conclusion section a bit so that he text is more specific (highlighted in yellow).

Reviewer 2 Report

The manuscript by Mooibroek reports an analysis of carbon-bonding of methyl groups in crystal structures from the Cambridge Structural Database (CSD) and the Protein Data Bank (PDB) utilizing Density Functional Theory (DFT) calculations. The survey of structures from the CSD and PDB coupled with the DFT study illustrates that these interactions are relatively weak in neutral adducts but are considerably stronger in systems with electron withdrawing atoms or cation species, consistent with prior studies. Overall, this manuscript will be of general interest to the readership of Molecules. The points below should be addressed before this work is considered for publication.

1) The manuscript should specify the energy decomposition method used to dissect the total interaction energy into electrostatic, orbital, and dispersion components.  

2) The manuscript should provide additional statistical information about the carbon bonding surveys. Specifically, what were the total number of crystal structures from the CSD and the PDB that were analyzed for methyl carbon bonds? What percentage of these structures from the CSD and the PDB displayed carbon bonds?

3) What criteria were used for selecting crystal structures from the PDB for analysis of methyl carbon bonding? For the CSD, an R-factor </= 0.1 was used for selecting structures for analysis. The manuscript should clarify the criteria used for selecting protein and DNA crystal structure from the PDB.

Author Response

I would like to thank referee 2 for his/her time and effort to suggest improvements to this article. Below is a response to his/her point and mutations have been highlighted in yellow in the newly submitted manuscript.

vs 1: Many thanks for this improvement. The energy decomposition scheme I used comes from the ADF suite as described in what are now references 29 and 31. This scheme has been used with success as detailed in reference 32 for hydrogen bonding. References 31 and 32 are new and highlighted in yellow, and the experimental section has a small addition to clarify this.

vs 2: These data are actually present in the paper. The total number of entries (and hits within these entries) from both the CSD and the PDB that were scrutinized are given in Table S1. Table S2 gives the amount of data for each dataset from Table S1, where angle α is 160 degrees or more. Table S2 also gives the number of hits within this dataset (of α ≥ 160 degrees) that display formal van der Waals overlap (in both absolute and relative terms). If one believes that structures compliant to these criteria are carbon bonds then Table S2 provides a numerical overview of their occurrence. The reason I do not emphasize these numerical data is because, in my opinion, compliance to geometric criteria alone is not direct evidence for the existence of an interaction. This is particularly true for very weak interactions, such as the ones described here. My preference therefor is to focus on distributions.

vs 3: As is stated in section 2.1: ‘The PDB was queried using Relibase[19] 3.2.3 and restricted to protein and DNA crystal structures where the packing environment was also searched. Datasets were obtained using the general query shown in Figure 1a.’. No other restrictions were applied because even high resolution protein structures are only high resolution on average and may contain rather flexible / imprecise regions. Reversibly, structures considered low resolution can contain very precise regions. A sentence is now added to specify that no other restrictions were imposed on the PDB search (highlighted in yellow).